# Use of Self-Figure Drawing as an Assessment Tool for Child Abuse: Differentiating between Sexual, Physical, and Emotional Abuse

**DOI:** 10.3390/children9060868

**Published:** 2022-06-11

**Authors:** Nisara Jaroenkajornkij, Rachel Lev-Wiesel, Bussakorn Binson

**Affiliations:** 1Faculty of Creative Arts Therapies, University of Haifa, 199 Aba Khoushy Av., Haifa 3498838, Israel; 2The Emili Sagol Research Center for Creative Arts Therapies, University of Haifa, Haifa 3498838, Israel; rlev@univ.haifa.ac.il; 3Social Work, Tel Hai College, Qiryat Shemona 1220800, Israel; 4Faculty of Fine and Applied Arts, Chulalongkorn University, Bangkok 10330, Thailand; bsumrongthong@yahoo.com

**Keywords:** child abuse, sexual abuse, physical abuse, emotional abuse, self-figure drawing

## Abstract

Child abuse is a worldwide phenomenon with adverse short- and long-term mental and physical negative consequences, with a huge gap between the prevalence of child abuse and disclosure rates. The study aimed to examine and validate the self-figure drawing as an assessment tool to differentiate between three forms of child abuse, i.e., child sexual abuse (CSA), child physical abuse (CPA), and child emotional abuse (CEA). Following the ethical approval, 1707 Thai children (13–18 years old) from the general population (schools) were asked to complete a self-report anonymous questionnaire consisting of four measures (Demographics, Childhood Trauma Questionnaire (CTQ), The Medical Somatic Dissociation Questionnaire (MSDQ), and The Disclosure of Trauma Questionnaire (DTQ)). After completion, they were asked to draw themselves. There was a significantly positive link between the reluctance to disclose and the experience of abuse, indicating that the more severe the abuse the higher the reluctance to disclose. The findings broaden the knowledge of movement and symbols as representations of inner personal conflictual material. Additionally, it substantiates self-figure drawing as an assessment tool and assists practitioners in early child abuse detection.

## 1. Introduction

Child abuse is a worldwide epidemic, resulting in actual or potential harm to the child’s health, development, and dignity [1], that has a devastating variety of short- and long-term physical and mental negative consequences [2]. The long-term consequences contain impaired capacities for trust, intimacy, and sexuality, including a variety of chronic mental and physical health problems [3].

Although child abuse has been classified into four different forms— sexual abuse, physical abuse, psychological abuse, and neglect—the overlapping and the inter-relationships between them bring such difficulty in evaluating the impact of each form towards mental consequences and disclosure [4]. Several studies reveal that multi-types of child abuse often co-occur [5], with neglect accompanied by physical and emotional abuse found to be the most common occurrence [6]. Emotional abuse has been proposed as the core of multi-type abuse, resulting in negative consequences of most cases of abuse [7].

Even though disclosure can bring an end to the child abuse, a great number of survivors keep silence for reasons such as feelings of fear and shame, difficulty to articulate the experience, or are silenced by the perpetrator or his or her supporters [8,9]. Moreover, the impact of abusive experiences on the brain area often leads to dissociation and an inability to recount the traumatic experiences [10].

Under the above circumstances, these proved to be a great challenge for investigators and practitioners to develop techniques to aid victims’ willingness to disclose. The self-figure drawing was developed by Lev-Wiesel [11] from the draw-a-person test [12] to identify significant drawing indicators in human figure drawings of individuals with traumatic experiences as it can be used to assess specific psychological and emotional characteristics.

Thus, by differentiating and categorizing indicators presented within drawings into each type of abuse, the self-drawing figure can become a valuable tool to identify the different types of abuse and to evaluate the mental health of individuals.

### 1.1. Childhood Sexual Abuse (CSA)

This term describes the condition where the child is being subjected to any behavior of sexual intent or content by an adult or another child that is older than them. Therefore, CSA may range from fondling to rape, non-contact abuse, e.g., voyeurism, exhibitionism or unwanted sexual comments, sexual exploitation, or any other sexual assault form [13].

Survivors of CSA are at high risk of developing subsequent physical health symptoms, i.e., general health, gynecological pain, cardiopulmonary symptoms, and obesity [14]. CSA was found to be strongly associated with clinical levels of psychological distress, low self-esteem, PTSD, anxiety, depression [15], eating disorders [16], borderline personality disorder, alcohol and drug abuse, and delinquency [17].

### 1.2. Child Physical Abuse (CPA)

According to Al-Shail et al. [18], the term is stated as the intentional use of physical force that caused harm, or high risk of harm, to the child. This definition generally includes a large variety of types and degrees of physical force, e.g., shoving, hitting, slapping, shaking, throwing, punching, biting, burning, or kicking.

There is a high prevalence of head and orofacial injuries in CPA, with bruising as a common injury among children [19]. CPA is a risk factor for a range of long-term consequences, i.e., depression, anxiety, low self-esteem, PTSD, borderline personality disorder, suicidal behavior, and aggression [20]. Suffering from CPA also results in eating disorders, insecure or disorganized attachments, and the tendency to become a victim of intimate partner violence [21].

### 1.3. Child Emotional Abuse (CEA)

The term comprises verbal and nonverbal degrading, terrorizing, exploiting, corrupting, ignoring, and isolating, as well as hostility, rejection, and the prevention of needed stimuli and/or the denial of emotional responsiveness [22]. The interactions, as well as inappropriate emotional responses from a parent, are in repeated patterns and perceived as typical in the parent–child relationship.

A large array of evidence indicates that childhood emotional abuse has adverse detrimental short- and long-term negative consequences [23] that involve a wide range of physical and psychological issues, e.g., PTSD, anxiety, depression [24,25], low self-esteem, substance abuse, obesity, fatigue, poor general health, chronic pain conditions [26], sexual difficulties, and a higher risk of committing suicide or having suicidal thoughts [21].

### 1.4. Multi-Type Abuse

In many cases, the child is subjected to multi-type abuse, especially CEA with either CPA or CSA [27]. It has been indicated that exposure to multi-type childhood abuse is associated to higher levels of mental and physical negative consequences severity [21,28]. In young children and adolescents, exposure to multi-type childhood abuse is likely to result in low self-esteem [29], eating disorders [30], antisocial behaviors, and aggressive behavior [31].

### 1.5. Child Abuse and Brain

Several research found anatomical, functional, and neurohormonal changes in the brain as a result of child abuse. However, it seems that different types of abuse produce their own changes to the brain. Exposure to CPA and CSA led to hyperactivation, decrement of the amygdala volume, reduction of the hippocampus thickness, and low connectivity at the medial prefrontal cortex. These areas were associated with the reduction of IQ, high aggressiveness and impulsiveness, and difficulty in differentiation threatening situations [32]. An alteration in the myelination of the corpus callosum and the decreasing of gray matter at the fusiform gyrus and the medial occipital gyrus were induced by CEA. These changes produced cognitive alterations in learning and memory, elevation of anxiety and depression, interruption of emotional regulation, and alteration of emotional perception of situations and faces [33]. Furthermore, it was believed that suffering from child abuse might cause pain reduction as the developing sensory system and pathways that transmitted disgust and traumatic experiences decreased.

Previous studies discovered the link between CSA and the structural and functional changes in limbic and prefrontal brain regions such as the amygdala, the prefrontal cortex, the orbital frontal cortex, and the subgenual area. The reduced activity in the orbital frontal cortex linked to damaged emotional inhibition, social behavioral problems, and violations, whereas the disruption in the subgenual area induced mood disorders such as depression [34,35].

Thus, experiencing different types of child abuse, particularly CSA, acted as a moderator between MSDQ and CTQ. A high severity of abuse predicted high detrimental impacts on an individual [36], in relation to MSDQ and its subscales: somatization, depression symptoms, and dissociation [37].

The brain is responsible for high-level cognitive functions such as planning and organizational abilities, body language, hand movements, and eye-hand coordination, which is required for daily activities [38]. As vision could not be separated from motion, drawing was one of the effective tools used in order to express an individual’s perception of the world by employing the coordination of visual perception and fine motor control. [39]. The right and left hemispheres were both engaged in the process of visualizing and drawing in different areas [40]. The left hemisphere is involved in verbal and rational knowledge, information processing, and analyzing. Therefore, in the context of drawing, the left hemisphere played the key role in the understanding of rules and systems, resulting in a quality drawing through several drawing practices. On the other hand, the right hemisphere entailed relationships, patterns, and intuition, which resulted in creativity [41]. The unconsciousness and internal world appeared through micro movements; the process was uncontrollable and mostly went unnoticed [42]. For that reason, the use of self-figure drawings could be one of the excellent paths for an individual to identify the unconscious signals and to explore and successfully bridge the gap between the consciousness and the unconsciousness.

### 1.6. Disclosure of Child Abuse

According to Ullman [43], disclosure is referred to the telling anyone of the abuse, including formal or informal support sources, with voluntary victim and non-victim initiations. It is a unique process of each victimized individual that is likely to be influenced by the aspects of culture, religion, gender, and the abuse itself [44]. In addition, Foynes et al. [45] propose that children’s age, gender, relationship to the offender, fear of negative consequences, perceived responsibility for the abuse, and characteristics of the abusive event are all part of the disclosure process as well. Not only does reluctance to disclose expose the child to the risk of further victimization it also prevents him or her in receiving the appropriate treatment. In the cases of CSA and CEA, it is difficult to identify the abuse due to the lack of visible physical injury, although apparent symptoms of distress are presented [46].

Several studies found that the most common self-reported reason for non-disclosure is that many children considered their experiences of violence not to be serious enough to report. They do not recognize their experiences as abusive or view some of the abusive acts as a normal part of everyday life [47,48].

### 1.7. Self-Figure Drawing as an Assessment Tool

Self-figure drawing is one form of a well-known and frequently used projective drawing technique for assessment in a clinical setting. Machover [12] developed the Machover Draw-A-Person (DAP) test with the assumption that, within the drawing, the human figure is the subject and the paper is the representation of the environment surrounding the subject. Lev-Wiesel [11] has developed a version of the draw-a-person test called the self-figure drawing that forces the drawer to relate directly to oneself. As such, drawing has been broadly regarded as an additional assessment tool for assessing different personality traits and adverse experiences.

So far, concerning research on drawings of CSA, CPA, and CEA survivors, although there are no studies that compared these three abuses, the effects and consequences were studied. CPA children often exhibit tendencies of aggressiveness, which were found in indicators, e.g., the presence of straight lines, clawed fingers, long arms, and pressed lines [49]; shaded, crossed, hollowed, or omitted eyes; emphasized eyebrows; thick mustaches and beards; large pointed clawed fingers; broad shoulders and strong posture [50]; teeth, emphasized nostrils [51].

The study by Lev-Wiesel [52] presented four indicators of experiencing CSA, i.e., face line (double, hollow, or shaded), eyes (dots, omitted, shaded, hollow), hands and arms (clinging, cut off, detached, omitted), and genitals (shaded, blocked, disconnected from the rest of the body). Research by McInnes [53] found that within self-figure drawings, despite the absence of important facial details, the mouth is prominent. In the depiction of oral sex, the exaggerated mouths and teeth are often featured in drawings, while an emphasis on genitals frequently occurs in the drawings with missing arms or legs. The floating position of a figure, unconnected to the ground, is associated with a dissociation experience, implemented as a coping strategy during the abuse experience.

Indications of CEA in self-figure drawings are often depicted as small, faintly drawn, or positioned at the paper corners, indicating low self-esteem or low self-concept [54]. The lack of facial details in a figure represents the lack of voice and identity, while the lack of arms represents the feeling of helplessness [53]. Anxiety is one of the symptoms of distress indicating signs of emotional abuse [24]. The indicators include face shading, broken/varying line, and off-balance figures reflecting insecurity (implying a state of anxiety) [55].

### 1.8. Thai Culture

Hierarchy is essential for Thai families, i.e., a child must obey their parents. This belief leads to the acceptance of the parent’s right to discipline their children and to the positive perception of physical punishment as a way for parents to express love to their children and not as abusive behaviors [56].

Many Thai parents do not perceive the importance and need to communicate with children about sex. They may lack the knowledge and skills about sex, are embarrassed to discuss sex within a family or are reluctant to discuss sex as it might encourage children to try sexual intercourse [57].

A study by Binson and Kinear [58] found that in self-figure drawings, Thai people tend to omit the legs. This may be due to their perception of feet as the dirtiest part of the body. Moreover, smiling faces are frequently presented to show the eagerness to be genuinely nice, friendly, and polite in order to preserve a harmonious relationship with others.

Based on the above review indicating that (a) there is a huge gap between the prevalence of child abuse and disclosure rates, (b) there is a lack of assessment tools that indicate child abuse, (c) self-figure drawing enables children to express aversive experiences in a non-verbal way, the following hypotheses are made: (a) there are drawing indicator differences to specify each type of abuse; (b) for each type of abuse, drawing indicators follow the findings of previous studies; and (c) the result of DTQ is able to be used in explanation in case of the negative correlation between drawing indicators and the CTQ and MSDQ scores. Thus, the results of the study can support the indication that the self-figure drawing is a reliable assessment tool for predicting child abuse of each specific type.

## 2. Materials and Methods

### 2.1. Participants and Procedure

Following receiving the certificate of approval no. 001.1/64 with the exemption of informed consent from the Research Ethics Review Committee for Research Involving Human Research Participants, Group 1, Chulalongkorn University, Thailand, 1707 Thai children between the ages of 13 and 18 were recruited from schools with different socio-economic status through a convenience sampling method. The exemption was granted to maintain the participants’ anonymity and confidentiality and to prevent the concealment of any disclosure to the extent feasible.

A convenience sampling method was applied in the proposed study. The schools located around Samutprakarn, Bangkok, and Nakhon Pathom were approached through letters asking for permission. Upon receiving the approval from the schools, another letter was sent to all the students aged between 13 and 18, detailing the study’s aims and procedure. Those who were interested would participate in the study. Participants were anonymously asked to indicate which events they experienced before filling out the questionnaires. Confidentiality was ensured and participants had the right to withdraw from the study at any time and for any reason. Participants who felt stressed by the procedure were free to contact the researcher through the given email. In addition, free professional counseling information, such as helpline1323 and samaritansthai, was provided. Due to the study’s anonymous design, no tracking could be made on participants’ access to the mentioned counseling resource.

Participants were asked to complete an anonymous questionnaire that consisted of three measures: the Childhood Trauma Questionnaire (CTQ), the Medical Somatic Dissociation Questionnaire (MSDQ), and the Disclosure of Trauma Questionnaire (DTQ), in addition to self-figure drawing (Draw Yourself). All the measures were adapted to Thai culture through translation and back translation. For meaning accuracy, the measures were first translated to Thai by two Thai researchers and were translated back to English by the other two researchers. The study offered both paper and online questionnaires; therefore, the participants might or might not have had access to the internet. The authors have no relevant financial or non-financial interests to disclose.

### 2.2. Measures

#### 2.2.1. Demographic Questionnaire

The questionnaire contains three questions focusing on demographic data, i.e., age, gender, and ethnicity, used in order to collect background information.

#### 2.2.2. Childhood Trauma Questionnaire (CTQ)

The CTQ short-form [59] comprising 25 items plus the three validity items, producing a 28-item short form. The 25 items of the CTQ refer to lifelong abusive experiences and cover five types of abuse, i.e., CEA, CPA, CSA, emotional neglect, and physical neglect; in a five Likert scale format, ranging from 1 (never true) to 5 (very often true). The Cronbach Alpha reliability scores of the five scales were 0.84–0.89 for CEA, 0.81–0.86 for CPA, 0.92–0.95 for CSA, 0.85–0.91 for emotional neglect, and 0.61–0.78 for physical neglect.

#### 2.2.3. The Medical Somatic Dissociation Questionnaire (MSDQ)

The questionnaire was developed by four experts in child abuse and dissociative disorders [37]. The questionnaire includes 30 items, all of which were written in behavioral terms with no reference to the terms ‘somatic’ or ‘dissociation’. The items of the MSDQ covered three factors of somatization, depression symptoms, and dissociation. The questionnaire employs a 5-point Likert-type scale, ranging from 0 (nothing) to 4 (extremely). The Cronbach’s alpha for the full MSDQ was 0.93, and the Cronbach’s alpha of the three factors were 0.76 for somatization, 0.89 for depression symptoms, and 0.85 for dissociation. The MSDQ can differentiate between CSA survivors and the general population.

#### 2.2.4. The Disclosure of Trauma Questionnaire (DTQ)

The questionnaire [60] focuses on aspects of an individual’s intention to disclose traumatic events. It employs a Likert scale ranging from 0 (not at all) to 5 (completely) for each of the questionnaire’s 34 items. The questionnaire comprises three subscales: reluctance to talk (13 items), assessing reported resistance to tell others about the trauma; urge to talk (11 items), assessing participants’ need to disclose traumatic experiences; and emotional reactions to disclosure (10 items), assessing affective states and experiences that may occur during disclosure. The Cronbach’s alpha reliability scores of the three DTQ scales were 0.82 for reluctance to talk, 0.88 for urge to talk, and 0.87 for emotional reactions.

#### 2.2.5. The Draw Yourself Test

Draw your self-figure, is a version of the draw-a-person test [12] developed by Lev-Wiesel [11]. Participants were asked to draw themselves on A4 paper with a pencil. No further instructions were given. Any question regarding the drawing was responded to with “as you wish”. The drawings had been given blindly to three professionals that assessed the level of obviousness (from very much obvious to not at all) of indicators found previously in the literature to indicate symptoms or traits of children who experienced CSA, CPA, and CEA.

### 2.3. Data Analysis

In this study, continuous variables were reported by means and standard deviations, while categorical variables were reported by frequencies and proportions. Associations between drawing indicators and CTQ, MSDQ, and DTQ scores were assessed using Spearman’s correlation coefficients. Univariate analysis was performed using chi-square or one-way ANOVA to test for association of the demographics and drawing indicators with groups of child abuse (CSA, CPA, and CEA). Post hoc pairwise comparisons adjusting for Tukey multiple testing were performed to compare between pairs of groups.

In order to test whether experiencing CSA, compared with experiencing CPA and CEA, to moderate the relationship between CTQ and MSDQ total scores, as well as subscales of DTQ and drawing indicators, PROCESS macro was used for model no. 1 as outlined by Hayes [61]. The same procedure was employed to test moderation of experiencing CSA compared with experiencing CPA.

The ROC technique was implemented to find an optimal cutoff of CTQ, MSDQ, and DTQ total scores that will best differentiate between subjects who experienced CSA compared with those with no experience of CSA. Additionally, the ROC technique was used to find an optimal cutoff that will determine the difference between subjects who experienced CPA compared with those who experienced CSA and CEA. These cutoff points were chosen by point maximizing the Youden function, which is the difference between sensitivity rate and specificity rate over all possible cut-point values [62]. All analysis was performed by SAS 9.4 for Windows.

## 3. Results

### 3.1. Characterizing Drawing Indicators and Demographical Differences by Groups of Abuse

Table 1 presents a comparison of demographics, drawing indicators, and scales’ means among CSA, CPA, and CEA participants. In the drawings of CPA participants, the hair indicator was more emphasized than those of CEA participants (MCPA = 2.17 vs. MCEA = 1.95, *p* = 0.005). CPA participants emphasized face line, ears, and hands and arms more than CEA participants (face line: MCPA = 2.30 vs. MCEA = 1.91, *p* < 0.001; ears: MCPA = 2.47 vs. MCEA = 1.26, *p* < 0.001; hands and arms: MCPA = 2.12 vs. MCEA = 1.94, *p* = 0.029). Of the three types of abuse, CSA participants showed the most emphasis on the face line part (MCSA = 3.67 vs. MCPA = 2.30 and MCEA = 1.91, *p* < 0.001). Concerning the ear indicator, there is a greater emphasis in the drawings of CSA participants compared with those of CEA participants. However, when compared with the drawings of CPA participants, those of CSA participants showed less emphasis (MCSA = 2.01 vs. MCEA = 1.26 and MCPA = 2.47, *p* < 0.001). CSA participants scored higher on the genital indicator than CPA and CEA participants (MCSA = 1.05 vs. MCPA = 1.00 and MCEA = 1.00, *p* < 0.001).

The MSDQ scale and the DTQ subscales (i.e., reluctance to talk and emotional reactions) are significantly lower for CEA participants than for CPA or CSA participants. However, there is no statistical differences between CSA and CPA participants. The CTQ scale is significantly higher for CSA participants compared with CPA and CEA participants. Moreover, the CTQ scale is also significantly higher for CPA participants compared with CEA participants.

### 3.2. Correlation of Drawing Indicators and CTQ, MSDQ, and DTQ Scales for All the Sample

Table 1 reports the Spearman’s correlation coefficients between CTQ, MSDQ, and DTQ scores and drawing indicators for all the sample. Correlation coefficients are all very low and smaller than 0.1. Due to the large sample size, correlation coefficients of absolute value of 0.08 are significant at 0.001 value. Reluctance to talk is positively correlated (r = 0.08–0.10) with hair, face line, and legs and/or feet (omitted, cut off). Emotional reactions are negatively correlated with arm position (r = −0.09). DTQ total score is positively correlated with legs and/or feet (omitted, cut off) (r = 0.09). Furthermore, depression is negatively correlated with arm position (r = −0.09), while dissociation is negatively correlated with nose (r = −0.08).

### 3.3. Experiencing CSA as Moderator of the Relation between CTQ, MSDQ, and DTQ Scales and Drawing Indicators

Experiencing CSA moderated the relationship between the MSDQ total score, the CTQ total score, the reluctance to talk score and urge to talk score, and the genital indicator (MSDQ: ∆R^2^ = 0.04, F(1, 1703) = 64.2, *p* < 0.001; CTQ: ∆R^2^ = 0.01, F(1, 1703) = 19.75, *p* < 0.001; reluctance to talk: ∆R^2^ = 0.03, F(1, 1703) = 58.5, *p* < 0.001; urge to talk: ∆R^2^ = 0.04, F(1, 1703) = 74.5, *p* < 0.001).

As illustrated in Figure 1, the direction of the relationship between the MSDQ total score, the CTQ total score, and the urge to talk score and obviousness of the genital indicator is different for CSA participants compared with non-CSA participants. For CSA participants, higher scales’ values are associated with less obviousness of the genital indicator. The higher reluctance to talk score is associated with a higher rating of obviousness in the genital indicator for CSA participants. However, for non-CSA participants, there is no relation between the MSDQ total score, the CTQ total score and urge to talk score, and the obviousness of the genital indicator.

Experiencing CSA moderated the relationship between the urge to talk score and the hair indicator, (∆R^2^ = 0.005, F(1, 1703) = 7.6, *p* = 0.006). As illustrated in Figure 2, for CSA participants, a higher urge to talk score is associated with more obviousness of hair and ear indicators. For non-CSA participants, there is no relation between the urge to talk score and the obviousness of hair.

Experiencing CSA moderated the relationship between the CTQ total score and the ear, the omitted hand and arm, and the omitted leg and feet indicators, (ears: ∆R^2^ = 0.003, F(1, 1703) = 4.4, *p* = 0.036; omitted hand and arms: ∆R^2^ = 0.003, F(1, 1703) = 4.5, *p* = 0.035; omitted legs and feet: ∆R^2^ = 0.005, F(1, 1703) = 7.9, *p* = 0.005).

As illustrated in Figure 2, for non-CSA participants a higher CTQ total score is associated with a higher rating of obviousness in the ear indicator. Meanwhile, for CSA participants there is no relation between the CTQ total score and the obviousness of the ear indicator. In addition, for CSA participants a higher CTQ total score is associated with less obviousness of the omitted hand, arm, leg, and feet indicators. For non-CSA participants, there is no relation between the CTQ total score and the obviousness of the omitted limb indicator.

### 3.4. Experiencing CSA Compared with Experiencing CPA as Moderator of the Relation between CTQ, MSDQ, and DTQ Scales and Drawing Indicators

Results reveals the significant and directional relation of the MSDQ total score, the CTQ total score, the reluctance to talk score and the urge to talk score, and the genital indicator. For CSA participants, higher scores of MSDQ, CTQ, and urge to talk are associated with less obviousness of the genital indicator. Meanwhile, a higher score of reluctance to talk is associated with more obviousness of the genital indicator.

For CPA participants, there is no relation between the MSDQ total score, the CTQ total score, the reluctance to talk score and urge to talk score, and the obviousness of the genital indicator. Additionally, regarding the significance and direction of the relation of the urge to talk score and the hair indicator, a higher MSDQ total score, CTQ total score, reluctance to talk score and urge to talk score are associated with more obviousness of the hair indicator for CSA participants. However, there is no relation between the MSDQ total score, the CTQ total score, the reluctance to talk score and the urge to talk score, and the obviousness of the hair indicator for CPA participants. Furthermore, considering the significance and direction of the relation of CTQ and the omitted leg and feet indicator, a higher MSDQ total score, CTQ total score, reluctance to talk score and urge to talk score are associated with less obviousness of the omitted leg and feet indicator for CSA participants. As for CPA participants, there is no relation between the MSDQ total score, the CTQ total score, the reluctance to talk score and the urge to talk score, and the obviousness of the omitted limb indicator. There is a significantly negative relation between the MSDQ total score and the obviousness of the ear indicator for CPA participants, while there is no such relation for CSA participants. Additionally, a higher MSDQ total score is associated with less obviousness of the ear indicator for CPA participants (∆R^2^ = 0.008, F(1, 719) = 5.9, *p* = 0.015).

### 3.5. Cutoff Total Scores for Sexual Abuse, N = 1707

A cutoff value of CTQ, MSDQ, and DTQ total scores differentiating CSA participants from non-CSA participants in all the sample was examined. When the CTQ score ≥ 2.1, the true positive rate (TPR or sensitivity: predicted as sexual abuse and they are truly sexual abused) is 67.5% while the 1-specificity (false positive rate, FPR: predicted as sexual abuse but they are not sexually abused) is 28.3%. When the MSDQ total score ≥ 2.2, the TPR is 73.0%, while the FPR is 48.9%. When the DTQ score ≥ 98, the TPR is 58.9%, while the FPR is 40.7%.

### 3.6. Cutoff Total Scores for Physical Abuse Compared with Emotional Abuse, N = 1544

After excluding CSA participants, a cutoff value of CTQ, MSDQ, and DTQ total scores differentiating CPA participants from CEA participants was examined. When the CTQ score ≥ 1.96, the TPR is 71.6%, while the FPR is 26.0%. When the MSDQ score ≥ 2.13, the TPR is 68.9%, while the FPR is 43.8%. When the DTQ score ≥ 98, the TPR is 48.4%, while the FPR is 36.4%.

## 4. Discussion

The current study investigated the differences in the drawings of Thai children aged 13–18 with different types of abuse (sexual, physical, and emotional abuse). The aim of the current study was to examine to what extent self-figure drawing can serve as a tool for assessing child abuse victimization. Specifically, the aim of this study was to investigate whether self-figure drawing can differentiate between forms of abuse. The three measures (CTQ, MSDQ, and DTQ) were also implemented in order to compare the scores among the three types of abuse. The findings indicated significant differences in the drawings that were associated with CTQ, MSDQ, and DTQ measures. Consistent with previous studies, different drawing indicators were found among groups of people who reported CSA, CPA, and CEA.

### 4.1. Drawing Indicators

In most cases, drawings offered symbols of the experienced abuse and/or depicted the abuse of an individual. The study examined the self-figure drawing in Thai culture, with a comparison to Israeli culture as the drawing indicators were based on the study with Israeli participants [63,64,65]. There were no differences found between the correlations between MSDQ and drawings in the Israeli, Thai, and Indian participants [66]; significantly, the emphasized face lines that were presented in the drawings of CSA participants compared with those of CPA and CEA participants (see Figure 3). This could denote the predicament in deciding between hiding or disclosing the traumatized experience as the emphasized face and the double cheeks or chin symbolized the willingness to vomit or swallow the said experience [11], while the pressure lines in drawing reflect the muscle tension and the energy level of an individual [67]. Genitals were also significantly presented in the drawings of CSA participants, which could be a positive indicator of sexual abuse [68], especially when combined with the presence of face line (double, hollow, shaded), eyes (dots, shaded, hollow, crossed, omitted), and hands and arms (clinging, detached, shadowed, omitted, cutoff) [69].

Interestingly, as can be seen in Figure 4, the current study found the significant indicators for CPA that are in accordance with the research by Lev-Wiesel et al. [63] in the following indicators: the emphasized or hair standing up (signifying insecurity, helplessness, fear, and anxiety), the emphasized or double ears (symbolizing the ability to receive and deal with the external world) [64,65], and the emphasized hands and arms (indicating helplessness and anxiety in dealing and interacting with the environment or the external world) [67]. The emphasized hair could also act as an indicator to the frontal lobe, suggesting the unconscious feelings of the uncontrolled impulsivity [64,70] or the brain injury as a result of the violence inflicted [71]. In this study, it was found that the emphasized or double face line was significant for the CPA group as well.

Contrary to the research by Lev-Wiesel [72], the shadowed eyes (indicating hiding feelings, suspicion of others, anxiety, fear of seeing and being seen), the omission of arms and hands (symbolizing anxiety, helplessness, and fearfulness), and the hair stand indicators did not present significantly more in the drawings of CEA participants when compared with those of CPA and CSA participants in the current study (see Figure 5). This might be due to the fact that in many cases of child abuse there was more than one type of abuse occurring [73]. Thus, this could be the result of the multi-type abuse where CEA co-occurred with CPA and CSA. Furthermore, another explanation could stem from the cultural differences between the Israelis and the Thais. Some behaviors, i.e., criticism and parents’ humiliation of children, appeared to be less considered as an emotional abuse in Thai culture compared with Israeli western culture. Contrastingly, in Thai culture, children were often raised to respect and adore their parents and were obligated to their parents regardless of how they were treated, while in Israel parents were expected to adore their children and were obligated to help their children for life [74].

### 4.2. The Association between CTQ and MSDQ

The current study showed the positive association between CTQ and MSDQ. A higher CTQ score indicated a higher MSDQ, suggesting that the child abuse greatly affected an individual. The severity of abuse resulted in higher psychological stress, greater functional disability, and poorer psychological adjustment, as well as increasing the risks of developing internalizing behavior problems (i.e., anxiety, depression, PTSD) and externalizing behavior problems (i.e., physical, verbal aggression, and disruptive behaviors) [75].

MSDQ and its subscales (i.e., somatization, depression symptoms, and dissociation) were constructed to aid the assessment process of physiological symptoms such as chronic pain [37]. Child abuse, especially CSA, played a key role in the manifestation of dissociation [76]. Dissociation is a mental process where there is a disconnection or a separation from thoughts, feelings, memories, and surroundings that affect an individual’s sense of identity and perception of time [77]. The current study found a negative correlation between dissociation and the nose indicator (indicating feelings of insecurity, inadequacy, and inferiority). Nostrils emphasized along with the nose suggested aggression and the impulse to act [44], which seemed to be in accordance with how child maltreatment associated with the development of aggressive behaviors [75].

Depression was known to be one of the common consequences of child abuse that extended to adulthood [78]. In this study, there was a negative correlation between depression and asymmetry or horizontal arm position (signifying crying for help, poor self-esteem, and avoiding contact with environment). The impact of early traumatic experiences on poor self-esteem could lead to the inclination of developing depression [79].

### 4.3. Disclosure of Abuse

Although the experience of child abuse and its short- and long-term effects may have been the topic of numerous studies and research, there was still a distinct difference between occurrences and disclosure rates [80]. Several risks factors had prevented and delayed a CSA individual to reveal what had happened to them, including fear of negative consequences of a disclosure for oneself and the perpetrator as CSA often took place within families or socially close relationships, fear of not being believed, fear of their parents’ reaction [81], difficulty in communicating the experiences [75], and the severity of the abuse [53]. In line with previous studies, the current study showed a significantly positive link between the reluctance to disclose and the experience of abuse, indicating that the more severe the abuse the higher the reluctance to disclose.

## Figures and Tables

**Figure 1 children-09-00868-f001:**
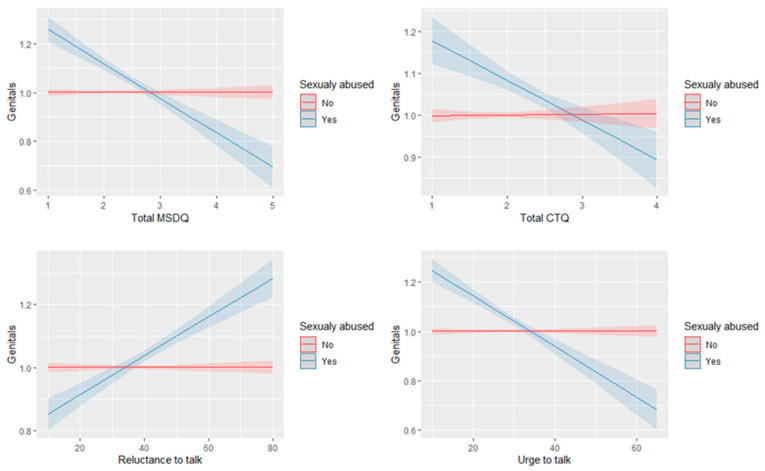
Visualization of significant interaction effect between experiencing CSA and MSDQ total score, CTQ total score, reluctance to talk score and urge to talk score, and the genital indicator.

**Figure 2 children-09-00868-f002:**
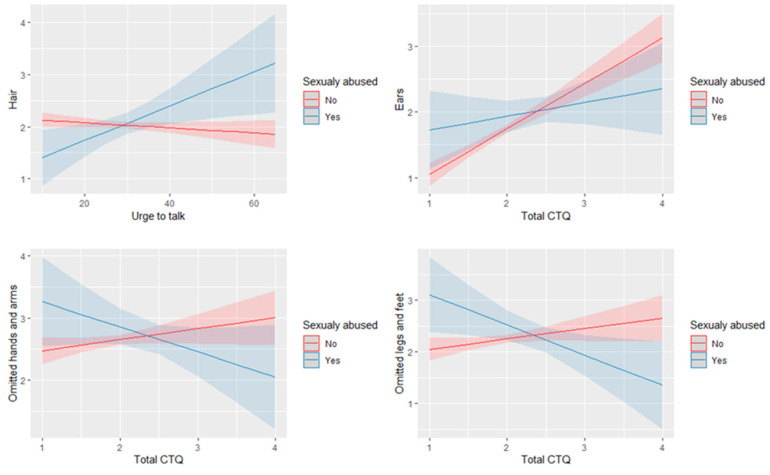
Visualization of significant interaction effect between experiencing CSA and urge to talk score or CTQ total score and the hair, the ear, the omitted hand and arm, and the omitted leg and feet indicators.

**Figure 3 children-09-00868-f003:**
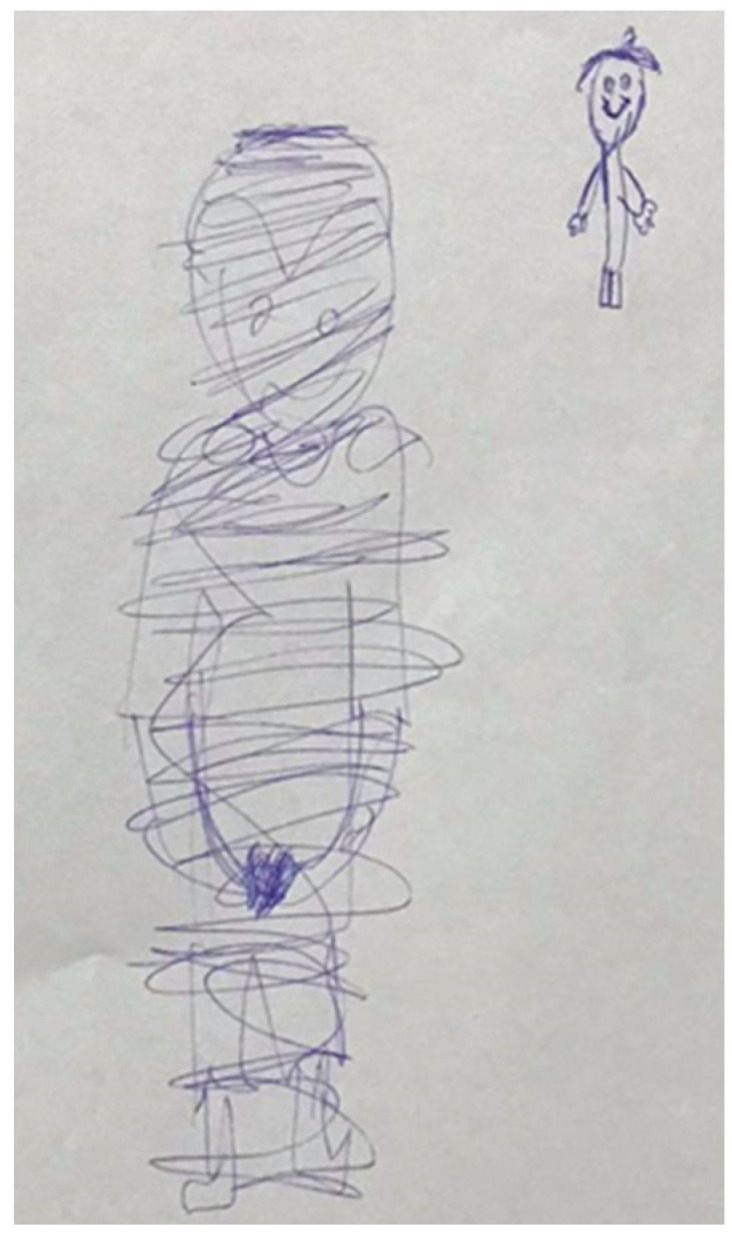
The drawing of a 16-year-old male participant who reported experiencing CSA. The significant drawing indicators included the presence of genitals and the emphasized face line.

**Figure 4 children-09-00868-f004:**
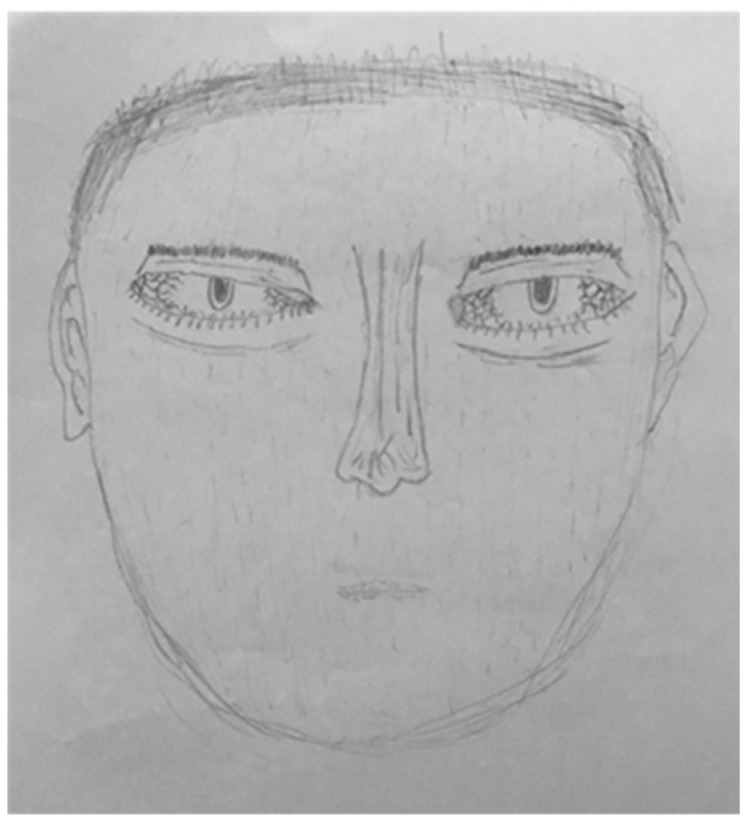
The drawing of a 16-year-old male participant who reported experiencing CPA. The significant drawing indicators included the emphasized or hair standing up, the emphasized or double ears, and the emphasized or double face line.

**Figure 5 children-09-00868-f005:**
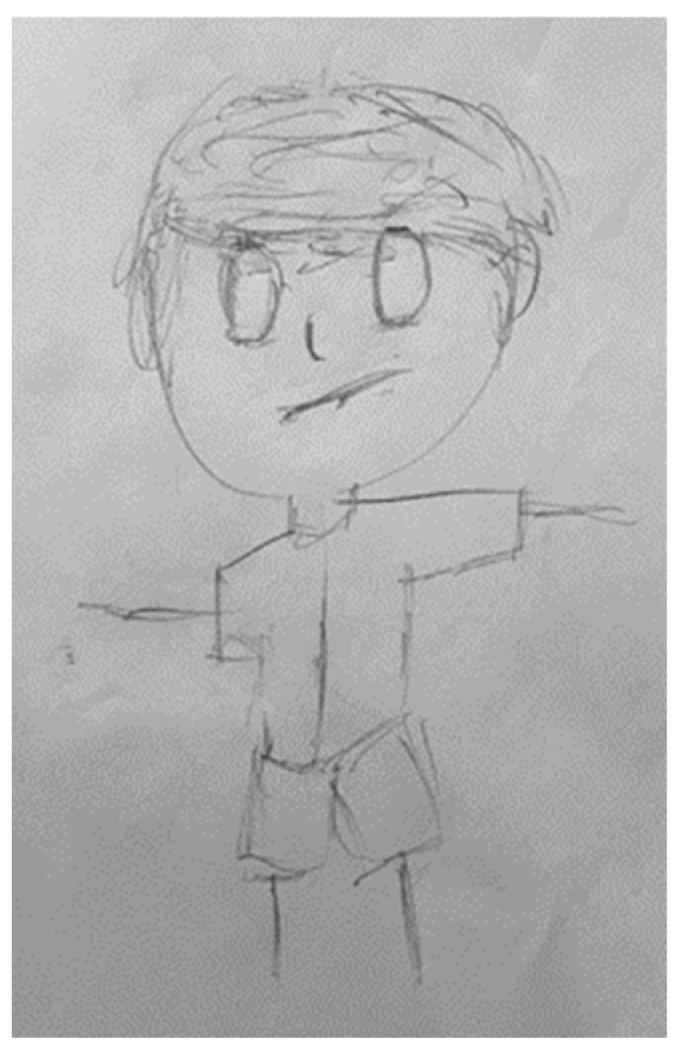
The drawing of a 15-year-old male participant who reported experiencing CEA. The drawing indicators found in the drawing included the hollow eyes, the omission of arms and hands, and the emphasized hair indicators.

**Table 1 children-09-00868-t001:** Characterizing Drawing Indicators and Demographical Differences Across Groups.

	CEA N = 984	CPA N = 560	CSA N = 163	*p* Value
Head (disproportionate size)	2.10 (1.27)	2.10 (1.29)	2.07 (1.29)	0.973
Forehead (emphasized, shadowed)	1.05 (0.36)	1.06 (0.39)	1.07 (0.42)	0.826
Hair (stand, emphasized)	1.95 (1.26)	2.17 (1.34)	2.04 (1.34)	0.005
Face line (double, hollow, shaded)	1.91 (1.23)	2.30 (1.34)	3.67 (0.52)	<0.001
Ears (emphasized, shadowed, double)	1.26 (0.77)	2.47 (1.47)	2.01 (1.38)	<0.001
Eyebrows (emphasized)	1.40 (0.95)	1.50 (1.04)	1.52 (1.10)	0.082
Eyes (dots, shaded, hollow, crossed)	3.36 (1.22)	3.38 (1.20)	3.40 (1.20)	0.923
Eyes (omitted)	1.11 (0.56)	1.16 (0.66)	1.17 (0.69)	0.261
Nose (emphasized, big, shadowed, nostrils)	1.57 (1.04)	1.61 (1.07)	1.64 (1.13)	0.581
Teeth (presence)	1.17 (0.68)	1.17 (0.68)	1.19 (0.72)	0.919
Moustache/beard (thick, shadowed)	1.03 (0.29)	1.05 (0.38)	1.02 (0.23)	0.346
Shoulders (broad)	1.72 (1.14)	1.77 (1.19)	1.80 (1.17)	0.621
Arm position (asymmetry, horizontal)	1.52 (1.02)	1.54 (1.04)	1.53 (1.06)	0.931
Hands/arms (clinging, detached, shadowed)	1.94 (1.35)	2.12 (1.45)	2.12 (1.38)	0.029
Hands/arms (omitted, cut off)	2.67 (1.45)	2.59 (1.45)	2.71 (1.45)	0.508
Fingers (large, pointed, clawed)	1.09 (0.47)	1.10 (0.50)	1.12 (0.57)	0.627
Genitals (shaded, blocked, disconnected)	1.00 (0.00)	1.00 (0.04)	1.05 (0.37)	<0.001
Legs/feet (distorted, disproportionate)	1.46 (0.72)	1.43 (0.70)	1.48 (0.76)	0.688
Legs and/or feet (omitted, cut off)	2.22 (1.47)	2.28 (1.47)	2.31 (1.48)	0.646
Posture (strong, stable)	1.15 (0.55)	1.16 (0.59)	1.07 (0.39)	0.145
Reluctance to talk	36.6 (11.8)	40.0 (12.2)	41.8 (11.6)	<0.001
Urge to talk	28.8 (9.18)	29.5 (8.53)	29.3 (7.72)	0.339
Emotional reactions	25.1 (8.80)	28.3 (10.3)	30.5 (9.61)	<0.001
DTQ total score	90.4 (22.9)	97.8 (25.2)	102 (22.4)	<0.001
MSDQ total score	2.10 (0.51)	2.41 (0.57)	2.48 (0.53)	<0.001
Somatization	2.10 (0.65)	2.45 (0.72)	2.53 (0.72)	<0.001
Depression	2.66 (0.74)	3.00 (0.76)	3.09 (0.71)	<0.001
Dissociation	1.66 (0.45)	1.93 (0.55)	1.98 (0.54)	<0.001
CTQ total score	1.80 (0.26)	2.17 (0.35)	2.36 (0.45)	<0.001

## Data Availability

The data are not publicly available due to ethical restrictions.

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
