# Peer review of "Use of Self-Figure Drawing as an Assessment Tool for Child Abuse: Differentiating between Sexual, Physical, and Emotional Abuse"

_children, 2022, doi:10.3390/children9060868_

Round 1

Reviewer 1 Report

The aim of the study is to examine and validate the self-figure drawing as an assessment tool to differentiate between three forms of child abuse: child sexual abuse (CSA), child physical abuse (CPA) and child emotional abuse (CEA).

I think the manuscript is well structured and well written. The abstract clearly reflects the objectives of the studies and the main findings. The chosen research methods are appropriate and correspond to the research questions. The literature review and hypothesis development are relevant.

The study is an attempt to rethink the expert value of  representations of inner personal conflictual material in self-figure drawing as a tool for assessing the early detection of child abuse based on correlation with the results of Childhood Trauma Questionnaire (CTQ), The Medical Somatic Dissociation Questionnaire (MSDQ), and The Disclosure of Trauma Questionnaire (DTQ). The study investigates the differences in the drawings of Thai children ages 13-18 with different types of abuse (sexual, physical, and emotional abuse). 

The article is precisely framed with well-defined statistical methods and also includes a very interesting conclusions.

I believe the article will provide adequate support for the future research of conceptualization and methodology of self-figure drawing as an assessment tool in early child abuse detection. 

I accept the article for publication in its current form.

Author Response

Dear Reviewer,

Thank you for your very careful review of my manuscript entitled “Use of Self-Figure Drawing as an Assessment Tool for Child Abuse: Differentiating Between Sexual, Physical and Emotional Abuse”, and for the comments, corrections and suggestions that ensued. A revision of the paper has been carried out to take all of them into account. Those comments are all valuable and very helpful for revising and improving the manuscript, and I hope the corrections meet with your approval.

Thank you and best regards.

Your sincerely,

Nisara Jaroenkajornkij

Reviewer 2 Report

Dear authors, dear editor

Thanks for letting me revise this interesting study.

I’ve really found some parts of the study engaging, while I got lost in other parts.

If you want to increase the dissemination of your article and give something meaningful to readers and inspiring for colleagues, you should, first of all, decide what is your aim and what is your audience.

Either you can adopt a more synthetical and technical description of you work (audience: experts in your field, methodologists), or a more narrative and descriptive version (audience: general professionals who can get inspiration and clinical applications reading your article). 

Moreover, it is not clear if the questionnaires you used were already cross-culturally adapted (if so, you can avoid reporting again here all the measures, if these is the first time they are used in your context probably it is better to dedicate the manuscript about the cross-cultural adaptation process).

Even, you can think about writing 2 different manuscripts about your work, one more academical and one more clinical. One dedicated to the cross-cultural adaptation of the questionnaires and one to disseminate the results of your study. Or to write just one manuscript, but clarifying the 2 different aims at the beginning, and creating an appendix were you include all the parts, more technical related to the cross-adaptation process .

Suggestions for the 2 versions. 

Introduction

Expert audience/ Cross-cultural adaptation of questionnaires

Summarize all the introduction in a few lines (in this case max 20 lines), providing

-        a few words about the overall topic of your research, including just the 1-2 references really important

-        a synthetic presentation of the previous research in the area (3-4 references)

-        the problem with the past research, the gap you want to cover in your study: the questionnaires you think are useful for studying this topic were never adapted to your context

-        what you did to fix that problem: you followed all the steps of the cross-cultural adaptation process

General audience/exploration of the use of drawings in the Thai culture regarding abuse 

In this case, you can keep the very good review of the literature you did and just clarify the aim of the study, revising and rewording the lines 139-147.

In these lines:

1. summarize the background of your topic and what comes from previous research in the area

2. what is the problem with past research, what is missing, what is the gap you are facing

3. what you did to fix that problem

Materials and Methods

Present in separate paragraphs

- the participants and the process (clearly, if it is a cross-cultural adaptation, the process should describe all the steps, better to provide a schema and clarify who translated, who back-translated and so on)

- the ethical concerns and comments in a separated paragraph (it could be at the end of the Materials and Methods section), including lines 150-152, and 154-156 and giving more space to the explanation of the exemption of the informed consent.

Results

If your manuscript is about a cross-cultural adaptation process it should include the final texts of the questionnaire, the different measurements related to the process and so on

If your manuscript aims to present the results of previously adapted questionnaires, please remove all the measurements and give more space to examples and figures, describing them in details.

Discussion

Write your discussion in accordance with your aim and your results. Again, clarify what is your aim before moving on.

Moreover:

Line 377 Contrary to the previous study [53],

This is the first time you mention this study, please specify it is written by you? Why are you mentioning it and using it in your discussion?

Line 420 Child Abuse and Brain

Even if really interesting this part doesn’t related directly to your study, either you expand it explaning why you think is important here, or you delete it.

Author Response

(The authors gave the same response as above.)

Reviewer 3 Report

I would like to congratulate the authors for the good work done.
I found it very interesting to read.
It is an original research, very well posed methodologically and very well described all sections.

I think it is very complete as it is for publication. I have nothing to contribute but congratulations.

Author Response

(The authors gave the same response as above.)

Round 2

Reviewer 2 Report

Dear authors

Thanks for having considered my suggestions and comments.

I think the manuscript is really interesting and helpful.

Another suggestion: remove the words race and ethnicity from your manuscript (line 146). I feel that culture, religion, and gender are enough for explaining the idea. Moreover, race and ethnicity are slippery concepts. 

Author Response

Dear Reviewer 2,

Thank you for your suggestion. I agree with your assessment, I have removed the words 'race' and 'ethnicity' (line 146) from the manuscript.

Kindly review the latest revised manuscript.

Please let me know if there's any further points of improvement or whether you think it's ready for publishing as is.

Thank you again for your time and feedback.

Wishing you a lovely week ahead.

Best regards,

Nisara Jaroenkajornkij